# Target-oriented Proactive Dialogue Systems with Personalization: Problem Formulation and Dataset Curation

**Jian Wang,   Yi Cheng,   Dongding Lin,   Chak Tou Leong,   Wenjie Li**

Department of Computing, The Hong Kong Polytechnic University

{jian-dylan.wang, alyssa.cheng, dongding88.lin, chak-tou.leong}
@connect.polyu.hk
cswjli@comp.polyu.edu.hk

## Abstract

Target-oriented dialogue systems, designed to proactively steer conversations toward predefined targets or accomplish specific system-side goals, are an exciting area in conversational AI. In this work, by formulating a <dialogue act, topic> pair as the conversation target, we explore a novel problem of personalized target-oriented dialogue by considering personalization during the target accomplishment process. However, there remains an emergent need for high-quality datasets, and building one from scratch requires tremendous human effort. To address this, we propose an automatic dataset curation framework using a role-playing approach. Based on this framework, we construct a large-scale personalized target-oriented dialogue dataset, **TOPDIAL**[1], which comprises about 18K multi-turn dialogues. The experimental results show that this dataset is of high quality and could contribute to exploring personalized target-oriented dialogue.

## 1 Introduction

Compared with traditional dialogue systems that focus merely on passively responding to user requirements, a recently investigated research topic of target-oriented dialogue systems (Sevegnani et al., 2021; Deng et al., 2023) specifies a conversation target from the system side, enabling the system to take the initiative and lead the conversation. Early work in this area mainly formulates the targets as mentioning certain keywords (Tang et al., 2019; Qin et al., 2020; Zhong et al., 2021; Yang et al., 2022) or specific topics (Wu et al., 2019; Sevegnani et al., 2021). To allow the formed targets to be applicable in broad scenarios, a few recent studies (Zhang et al., 2021; Wang et al., 2023b) define <dialogue act, topic> pairs as targets. For example, given the target of <movie recommendation, "King

of Comedy">, the system needs to take appropriate dialogue acts and smoothly steer the discussed topic towards the designated one. Its ultimate objective is to achieve recommendations on the target topic "King of Comedy". Our work also follows the form of <dialogue act, topic> pairs as targets to study target-oriented dialogue systems due to their higher applicability in real-world scenarios.

Despite many existing efforts, we find that two critical issues remain to be solved. One urgent problem is the need for well-organized benchmarks or datasets. Current studies for target-oriented dialogue (Gupta et al., 2022; Wang et al., 2023a) mainly re-purpose existing non-target-oriented dialogue datasets, which are not exactly suitable as they are crowd-sourced without consideration of target accomplishment. Nevertheless, building a new high-quality dataset from scratch requires expensive human effort. The other essential issue is that, target-oriented dialogue systems need to consider personalized aspects (Wu et al., 2021; Rana et al., 2023), such as user profiles and personalities, which were largely ignored by previous work. User profiles involve user preferences about potential topics relevant to the target, while personalities imply possible reactions and feedback during the dialogue process. With personalized information incorporated, the system could be tailored to a user and lead the conversation towards the target with higher engagement instead of obtrusively driving to the target, thereby improving user experience. Thus, we raise the question: *How can we build high-quality datasets with little human effort for personalized target-oriented dialogue?*

In this work, we first give a comprehensive definition (§2) of personalized target-oriented dialogue, then lay out the desirable characteristics (§2) that a qualified dialogue dataset should meet. Drawing inspiration from some recent work that has demonstrated unprecedented capabilities of large language models (LLM) in simulating human so-

---

[1]Our code and data are available at https://github.com/iwangjian/TopDial.

cial behaviors (Guo et al., 2023; Li et al., 2023), we propose a role-playing approach for automatic dataset curation (§3) using multiple LLM agents. They are designed to follow specific instructions to fulfill the requirements. Based on that, we synthesize a large-scale dialogue dataset named **TOP-DIAL** and show its quality and effectiveness (§4).

Our main contributions are: (1) We formulate the problem of personalized target-oriented dialogue, which is promising yet underexplored. (2) We propose a novel role-playing framework for automatic dialogue dataset curation. It provides insights into building large-scale datasets for many other dialogue tasks. (3) Our constructed TOPDIAL dataset is of high quality and contributes to the related research community.

## 2   Problem Formulation

**Task Definition**   We consider a dialogue corpus $\mathcal{D} = \{(\mathcal{U}_i, \mathcal{K}_i, \mathcal{T}_i, \mathcal{C}_i)\}_{i=1}^N$, where $N$ is the total number of dialogues. In the $i$-th dialogue, $\mathcal{U}_i$ represents the personalized information, such as the user's profiles and/or personalities. $\mathcal{K}_i$ represents the domain knowledge facts relevant to the $i$-th dialogue. $\mathcal{T}_i$ denotes the predefined target consisting of an <dialogue act, topic> pair. $\mathcal{C}_i = \{\mathcal{C}_{i,t}\}_{t=1}^{N_T}$ is the dialogue content, with a total of $N_T$ turns. The task of personalized target-oriented dialogue is formalized as follows: given a target $\mathcal{T}$, a set of user's personalized information $\mathcal{U}$, a set of relevant domain knowledge $\mathcal{K}$, and a dialogue context $\mathcal{C}$, the objective is to proactively lead the conversation and generate proper utterances to achieve the target $\mathcal{T}$ at an appropriate time.

**Desirable Characteristics of Datasets**   Based on the above definition, we lay out two desirable characteristics that a qualified dataset should meet, namely *target-oriented proactivity* and *personalization*. Target-oriented proactivity emphasizes that a dialogue dataset should allow the system to (i) take the initiative throughout a conversation, (ii) proactively lead the discussed topic towards the target topic based on domain knowledge, and (iii) accomplish the target act. On the other hand, personalization indicates that dialogues in a qualified dataset should embody (i) user profiles, which may involve users' past preferences about potential topics relevant to the target, and (ii) user personalities, which may imply users' possible reactions and feedback during the system-initiative process.

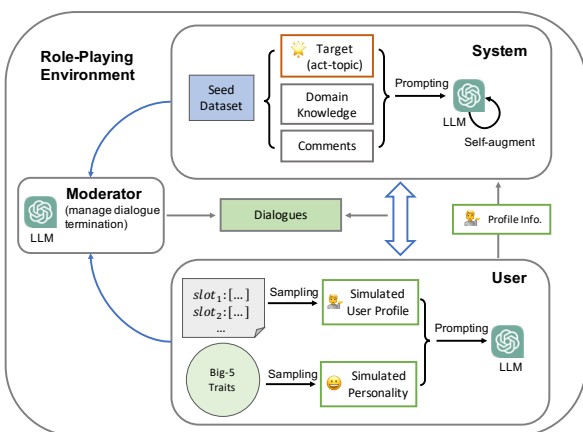

Figure 1: Overview of our role-playing framework for automatic dialogue dataset curation.

## 3   Dataset Curation Framework

In this section, we describe a role-playing approach for automatic dataset curation using multiple LLM agents. Figure 1 depicts the whole framework, which involves one *user agent*, one *system agent*, and one *moderator agent*. All these agents are designed to follow specific instructions and communicate in our *role-playing environment*.

**Role-Playing Environment**   This environment is designed to provide a global description for prompting all LLM agents. To achieve desirable target-oriented role playing, we instantiate the environment description based on the domains of the predefined targets. For example, one can describe the environment as "You are participating in a conversation about music or movies." for a given target $\mathcal{T}$ = <movie recommendation, "King of Comedy">. Then, the description will be prepended to each agent's instructions.

**User Agent**   The user agent aims to simulate human users who generate utterances conditioned on their specific profiles and personalities. Since there are many off-the-shelf dialogue datasets grounded with user profiles, we collect all user profiles from one chosen dataset and parse them into a profile slot pool. Each slot contains a particular slot key (e.g., name, age range, liked or disliked movies) and a list of candidate values. We randomly sample a slot value for each key, and then form all key-value pairs as the simulated user profile.

Inspired by Big-5 personality traits (Goldberg, 1993) that have been widely adopted in personality-aware tasks (Oraby et al., 2018; Yu et al., 2019), we randomly sample a positive or negative description

| Dataset | Participants | Formed Targets | TO | PF | PN | Domains | MT | #Dialogue |
|---|---|---|---|---|---|---|---|---|
| TGC (Tang et al., 2019) | Crowd workers | Keywords | ✓ | ✗ | ✗ | Open-domain | ✓ | 9,939 |
| DuConv (Wu et al., 2019) | Crowd workers | Topical entities | ✓ | ✗ | ✗ | Movies | ✓ | 29,858 |
| TG-ReDial (Zhou et al., 2020) | Crowd workers | N/A | ✗ | ✓ | ✗ | Movies | ✓ | 10,000 |
| OTTers (Sevegnani et al., 2021) | Crowd workers | Topics | ✓ | ✗ | ✗ | Open-domain | ✗ | 4,316 |
| TGConv (Yang et al., 2022) | Crowd workers | Keywords | ✓ | ✗ | ✗ | Open-domain | ✓ | 18,878 |
| DuRecDial 2.0 (Liu et al., 2021) | Crowd workers | N/A | ✗ | ✓ | ✗ | Movies, music, food, POIs[*] | ✓ | 16,482 |
| DuRecDial 2.0[†] (Wang et al., 2023a) | Human experts | Act-topic pairs | ✓ | ✓ | ✗ | Movies, music, food, POIs[*] | ✓ | 6,080 |
| TOPDIAL (Ours) | LLM agents | Act-topic pairs | ✓ | ✓ | ✓ | Movies, music, food, POIs[*] | ✓ | 18,009 |

Table 1: Comparison between TOPDIAL and other related datasets (TO: target-oriented, PF: profile grounding, PN: personality grounding, MT: multi-turn conversation, †: re-purposed version, *: point-of-interest restaurants).

for each of the following traits: openness (O), conscientiousness (C), extraversion (E), agreeableness (A), neuroticism (N). The sampled descriptions are then combined as the simulated user personality. We verbalize the simulated user profile and personality in natural languages, prompting the user agent to act as a human user. We present our detailed instruction template in Appendix A.1.

**System Agent** The system agent aims to serve as a human-like domain-specific enthusiast, such as a movie enthusiast who enjoys a variety of films, or a foodie who enjoys delicious food. Its long-term goal is to proactively lead the conversation towards the target, as discussed in §2. To achieve target-oriented proactivity, we take a given target $\mathcal{T}$ and a set of relevant domain knowledge $\mathcal{K}$ (and a few comments related to the target topic, if applicable) from a chosen seed dataset as the fundamental prompting source. Besides, in human-to-human conversations, one can easily know the other's explicit profile information, while it is hard to be aware of implicit personality before their first conversation. Thus, we pass the simulated user profile yielded by the user agent to the system agent as a personalized prompting source (see Figure 1).

We assign required instructions to the system agent based on the above prompting sources and task definition. We provide the instruction template in Appendix A.2. In practice, we further enhance the system agent in a self-augmented instruction manner, where the agent's task prompt will be repeated at each dialogue round to avoid forgetting its long-term goal.

**Moderator Agent** The moderator agent is designed to automatically manage the termination of the conversation between the system and the user agents. To ensure that the synthetic data adhere to desirable characteristics, we set certain conditions to terminate the conversation. These condi-

| | |
|---|---|
| Total # dialogues (train / valid / test) | 12,601 / 1,802 / 3,606 |
| Total # utterances (train / valid / test) | 141,928 / 20,310 / 40,496 |
| Total # targets | 501 |
| Avg. # slot keys per user profile | 10 |
| Avg. # traits per user personality | 5 |
| Avg. # knowledge triples per dialogue | 36.8 |
| Avg. # utterances per dialogue | 12.3 |
| Avg. # words per user's turn | 28.2 |
| Avg. # words per system's turn | 37.5 |

Table 2: Statistics of the TOPDIAL dataset.

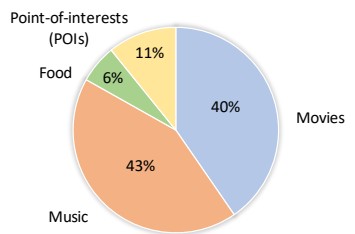

Figure 2: Distribution of domains among dialogues.

tions are outlined as follows: (1) The system agent completes the target act (e.g., recommendation) on the target topic, the user agent accepts it, and the system no longer takes the initiative for two rounds. (2) The user agent explicitly rejects the system agent's act on the target topic for the second time. (3) The conversation between the system and the user agents reaches a maximum number of rounds. For the first two conditions, we take a few dialogues from the seed dataset as in-context examples to demonstrate whether or not an ongoing conversation should be terminated. We present the detailed instruction template in Appendix A.3.

**Dataset Curation** We employ three ChatGPT (`gpt-3.5-turbo` version) agents as LLM agents for the above roles. We ask the system agent to initiate a greeting with the user agent, and they will chat turn by turn, resulting in multi-turn conversations. Their conversations are terminated by the moderator agent or the maximum limit of rounds.

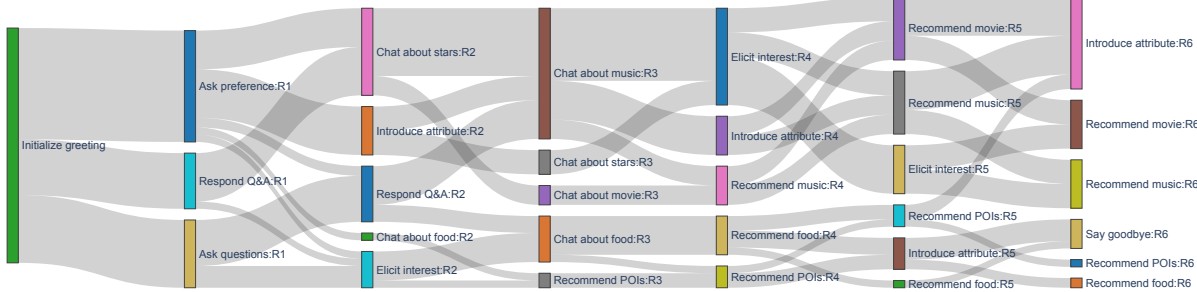

Figure 3: Transitions of dialogue acts of the system through the first six rounds.

The three agents can generate large-scale dialogues through their collaboration, with very little human effort involved in the whole process.

## 4 TOPDIAL Dataset

Based on our dataset curation framework, we synthesized the dataset **TOPDIAL** by utilizing the re-purposed version (Wang et al., 2023a) of DuRec-Dial 2.0 (Liu et al., 2021) as the seed dataset after carefully considering the problem formulation and necessary prompting sources. We report more implementation details in Appendix B.1.

**Dataset Statistics** Table 1 compares TOPDIAL with related datasets. To the best of our knowledge, TOPDIAL is the first dataset equipped with the desirable characteristics discussed in §2. It should be noted that the DuRecDial 2.0 dataset is crowd-sourced without considering targets and is not exactly suitable for the end task of target-oriented proactive dialogue, while the re-purposed version of DuRecDial 2.0 largely relies on human effort to form targets and preprocess dialogues. In comparison, our TOPDIAL dataset is curated based on target-oriented proactivity. In addition, by grounding the personality information during the dataset curation process, TOPDIAL is more natural and effective in reflecting personalization.

Table 2 shows detailed statistics of the TOPDIAL dataset (see domain distributions in Figure 2). We also visualize the transitions of dialogue acts of the system through the first six dialogue rounds in Figure 3. We observe that the system often asks preferences or other questions at the very beginning. As the dialogue continues, the system introduces topic-related attributes and elicits the user's interest. It shows that the system proactively leads the dialogue and gradually achieves target dialogue acts, i.e., recommendations on target topics.

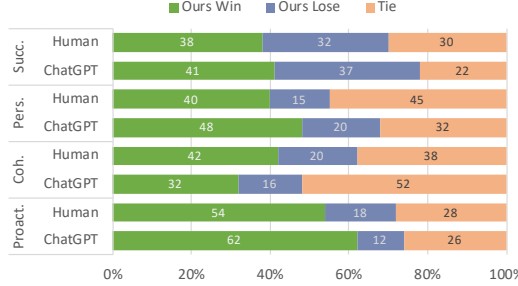

Figure 4: Automatic and human evaluation results between the seed dataset and ours (TOPDIAL).

**Automatic and Human Evaluations** To assess the quality of TOPDIAL, we conduct LLM-based automatic evaluation and human evaluation. We randomly choose 100 targets and then sample one dialogue per target from the seed and TOPDIAL datasets, respectively. We ask ChatGPT (OpenAI, 2022) and human evaluators to compare each pair of dialogues over four metrics: proactivity (Proact.), coherence (Coh.), personalization (Pers.), and target success rate (Succ.). We provide details for these metrics and our evaluation settings in Appendix B.2.

Figure 4 shows the evaluation results, where Fleiss's kappa (Fleiss, 1971) scores are distributed between [0.41, 0.60], indicating moderate inter-evaluator agreement. We observe that for all metrics, the TOPDIAL dataset achieves comparable and slightly higher win percentages over the seed dataset. It verifies the high quality of TOPDIAL.

**Dataset Evaluation by Baseline Models** We quantitatively evaluate TOPDIAL using representative dialogue models, including DialoGPT (Zhang et al., 2020) and Alpaca-7B (Taori et al., 2023). We fine-tune these models on the seed and TOPDIAL datasets, respectively, with an identical training data size. For a fair comparison, we build the test set for evaluation with 50% from the seed test data

| Model | Avg. BLEU | Knowledge F1 (%) | Persona F1 (%) | Succ. (%) |
|---|---|---|---|---|
| DialoGPT w/ S | 0.127 | 24.62 | 21.55 | 32.94 |
| DialoGPT w/ T | **0.138** | **47.42** | **30.51** | **51.83** |
| Alpaca-7B w/ S | 0.177 | 38.60 | 37.05 | 48.78 |
| Alpaca-7B w/ T | **0.229** | **57.12** | **51.99** | **85.04** |

Table 3: Performance of baseline models trained on the seed (S) dataset and our TOPDIAL (T) dataset.

and 50% from the TOPDIAL test data. Our evaluation metrics include the average score of BLEU-1/2 (Papineni et al., 2002), persona F1 (Lim et al., 2022), knowledge F1 and target success rate (Succ.) (Wang et al., 2023a). We describe details of these metrics and model training in Appendix C.

The comparison results reported in Table 3 show a similar trend: the two baseline models trained on our TOPDIAL dataset significantly outperform those trained on the seed dataset. In particular, our TOPDIAL dataset is more effective in training personalized target-oriented dialogue models (e.g., much higher persona F1 and Succ. socres) by grounding the profile and personality information during the dataset curation process. It shows that TOPDIAL is an effective training resource for the personalized target-oriented dialogue task.

**Case Study**   Due to space limitation, we present some cases in Appendix D (see Figure 9 and Figure 10) for a better understanding. These cases intuitively show that our TOPDIAL dataset fulfills target-oriented proactivity and personalization. It also shows that our dataset curation framework can be a viable alternative for building personalized target-oriented dialogue datasets.

## 5   Conclusion

In this work, we explore a new task: personalized target-oriented dialogue. We first define this challenging task, and then lay out the desirable characteristics that a qualified dialogue dataset should meet. We propose a novel role-playing framework for automatic dataset curation, based on which we construct a large-scale dialogue dataset TOPDIAL. Our statistics and evaluations validate its effectiveness and high quality.

## Limitations

Since we adopt ChatGPT agents to simulate the designed roles, ensuring the factual correctness of the synthetic dialogues during the role-playing process is challenging, as ChatGPT may produce output content with hallucinations (Bang et al., 2023). We intend to improve the dataset curation process with some post-processing steps, such as fact-checking and correction based on the grounded domain knowledge. In addition, we observe that sometimes the moderator agent cannot appropriately terminate a conversation due to its difficulty in understanding the achievement of the target, even though it has been assigned with detailed instructions and in-context examples. We will leave this for future research.

## Ethical Considerations

Developing target-oriented dialogue systems requires careful ethical considerations due to the potential impact on specific scenarios. As an application scenario explored in this work, providing recommendations is one of the highly-applicable target dialogue acts. Target-oriented dialogue systems can create non-obtrusive recommendations for specific products and services. Our work does not force the system to achieve the designated target nor force users to accept recommendations.

We emphasize that regulation of the target designation is crucial when deploying target-oriented dialogue systems in particular domains. For instance, specifying a target should not violate factual correctness, user privacy rules, or laws of human society. We want to raise awareness about the potential misuse of such systems with toxic intentions. For example, such systems may be used to pose as humans and mislead users through conversations. To avoid such risks, we highlight that it is necessary to improve transparency, such as informing users that they are chatting with a bot, not a human.

## Acknowledgments

This work was supported by the Research Grants Council of Hong Kong (15207122, 15207920, 15207821, 15204018, 15213323) and National Natural Science Foundation of China (62076212). It was also supported in part by PolyU internal grants (ZVQ0, ZVVX).

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

# A  Instructions for Different Agents

## A.1  User Agent

We provide the assigned instruction template for the user agent in Figure 5.

## A.2  System Agent

We provide the assigned instruction template for the system agent in Figure 6.

## A.3  Moderator Agent

We provide the assigned instruction template for the moderator agent in Figure 7.

# B  TOPDIAL Dataset

## B.1  Implementation Details of Dataset Curation

In this work, we implemented our role-playing framework based on an open-source library named ChatArena[2]. We called the gpt-3.5-turbo version of ChatGPT API[3] to build each LLM agent. We adopted a temperature of 0.75 to generate responses for all agents. We set the maximum number of tokens to generate to 100, 80, and 20 for

---

[2] https://github.com/chatarena/chatarena
[3] https://platform.openai.com/docs/api-reference/chat

**User Profile-specific Prompt:**

```
You are <USER_NAME>, a male/female student in the age range of <AGE_RANGE>, living in <RESIDENCE>
| a man/woman in the age range of <AGE_RANGE>, working in a company and living in <RESIDENCE> |
a retired man/woman in the age range of <AGE_RANGE>, living in <RESIDENCE>.
Based on your past experiences, you have the following preferences:
Your liked <SLOT_KEY>: <SLOT_VALUE>
...
Your disliked <SLOT_KEY>: <SLOT_VALUE>
...
```

**User Personality-specific Prompt:**

```
Based on the Big-5 personality traits, your personality is measured as:
For openness, you are (intellectual, imaginative, and curious | unimaginative, uncreative, and
conventional).
For conscientiousness, you are (efficient, organized, and careful | inefficient, careless, and
sloppy).
For extraversion, you are (outgoing, energetic, and talkative | shy, reserved, and quiet).
For agreeableness, you are (trustworthy, straightforward, and generous | unreliable, complicated,
meager, and boastful)
For neuroticism, you are (sensitive, nervous, and insecure | secure, confident, and calm).
```

**Task Prompt:**

```
Your response should be concise (no longer than 30 words).
You don't need to recommend anything, but feel free to express your personal interests.
You don't need to prepend your name to your response, despite others may do it.
```

Figure 5: Instruction template for the user agent. This involves the user profile-specific prompt, user personality-specific prompt, and task prompt.

**System Role Prompt:**

```
You are <SYSTEM_NAME>, a movie enthusiast who enjoys a variety of films | a music enthusiast
who enjoys a variety of music | a foodie who enjoys delicious food | a food enthusiast who is
interested in exploring different restaurants.
```

**User Profile-specific Prompt:**

```
You are conversing with <USER_NAME>, whose profile is below:
## <USER_PROFILE>
```

**Task Prompt:**

```
Your goal is to proactively lead the conversation with <USER_NAME> towards the target (movie |
music | food | point-of-interest, POI) <TARGET_TOPIC>.
To start the conversation, please begin with a greeting and avoid mentioning the target (movie
| music | food | POI).
As the conversation progresses, use your domain knowledge to steer the topic threads towards
the target (movie | music | food | POI) step by step.
Be informative and engaging while providing insights to arouse <USER_NAME>'s interest.
Remember to ultimately recommend <TARGET_TOPIC> as the focus of the conversation.
Your words at each turn should be concise (no longer than 30 words).

You may access the following domain knowledge for conversation:
## <DOMAIN_KNOWLEDGE_TRIPLES>
```

Figure 6: Instruction template for the system agent. This involves the system role prompt, user profile-specific prompt, and task prompt.

```
You are the moderator of a conversation. You need to determine whether the discussion between
<SYSTEM_NAME> and <USER_NAME> should come to an immediate end.
The conversation should be terminated under the following two conditions:
(1) If <SYSTEM_NAME> completes recommendation on <TARGET_TOPIC> and <USER_NAME> accepts it, and
<SYSTEM_NAME> no longer takes the initiative for two rounds.
(2) If <USER_NAME> explicitly rejects <SYSTEM_NAME>'s recommendation on <TARGET_TOPIC> when
<SYSTEM_NAME> has tried to recommend it for the second time.
In either of these cases, the conversation should be brought to an immediate end.

For example, here is a conversation: ## <SEED_DIALOGUE_1>
Should the conversation end? The answer is no.
Here is another conversation: ## <SEED_DIALOGUE_2>
Should the conversation end? The answer is yes.

Now, for the following conversation:
## <ONGOING_DIALOGUE>
Should the conversation end? Answer yes or no.
```

Figure 7: Instruction template for the moderator agent. This involves two comparative in-context examples to improve the instruction.

the system, user, and moderator agents, respectively. We set a maximum limit of 8 rounds based on our observation of target accomplishment while ensuring that the dataset curation is not too costly. We synthesized three different dialogue instances for each seed example in the chosen seed dataset, i.e., the repurposed version (Wang et al., 2023a) of DuRecDial 2.0 (Liu et al., 2021). On average, the cost of API calls is approximately 0.032 $ for one dialogue. We obtain two types of splits for the test set: *seen* and *unseen*, similar to Sevegnani et al. (2021); Wang et al. (2023a). The test-unseen split ensures that none of the target topics in the test set are present in the training set, whereas the test-seen split allows them to appear.

## B.2 Settings of Automatic and Human Evaluations

We describe the settings for LLM-based automatic evaluation and human evaluation that we conduct to validate the quality of the constructed TOP-DIAL dataset. We randomly choose 100 targets and then sample one dialogue per target from the seed and TOPDIAL datasets, respectively. We only include the targets and dialogue contexts while excluding grounded contexts (e.g., domain knowledge and personalized user information) for anonymity, since the grounded contexts of the seed and TOPDIAL datasets are distinguishable. For LLM-based automatic evaluation, we employ the `gpt-3.5-turbo` version of ChatGPT to compare each pair of dialogues. For human evaluation, we recruit three well-educated graduate students as

evaluators and ask them to perform a blind pairwise comparison. Specifically, we employ ACUTE-EVAL (Li et al., 2019), a widely used dialogue evaluation platform for multi-turn dialogue evaluation (Dinan et al., 2020; Kim et al., 2022). We adopt Fleiss's kappa (Fleiss, 1971) to measure the agreement among the human evaluators. Figure 8 shows the interface used for human evaluation.

We ask ChatGPT and human evaluators to compare each pair of dialogues in terms of the following metrics: proactivity (Proact.), coherence (Coh.), personalization (Pers.), and target success rate (Succ.), similar to related studies (Wang et al., 2023a; Kim et al., 2022). We use a question form to describe these metrics, with the wording of questions presented as follows:

- **Proactivity (Proact.)**: Which dialogue shows that the system takes the initiative during the conversation and proactively leads the topic threads toward the target topic?

- **Coherence (Coh.)**: Which dialogue is more natural and coherent, like humans? Whose dialogue context flows more smoothly?

- **Personalization (Pers.)**: Which dialogue reflects the user's preferences or personalities more? Which dialogue is more likely to arouse the user's interest?

- **Target Success Rate (Succ.)**: Which dialogue successfully achieves the target dialogue act on the target topic?

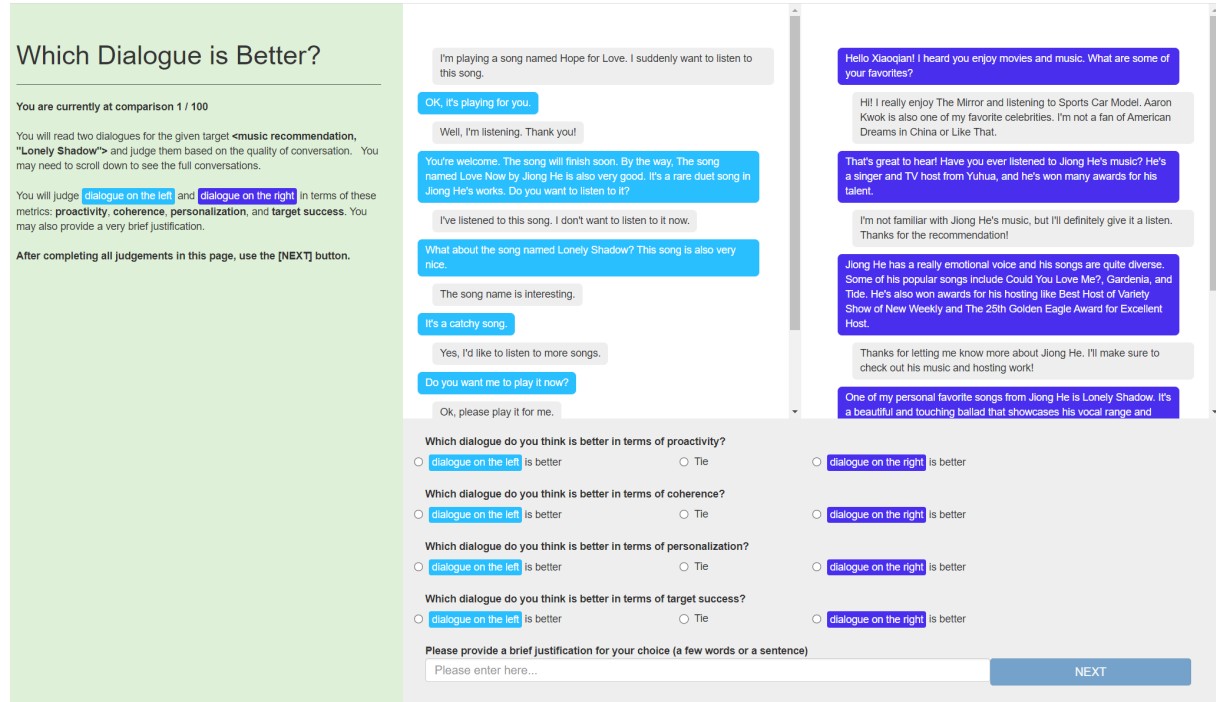

Figure 8: Interface for human evaluation. Here is a pair of dialogues from the seed dataset (left) and TOPDIAL dataset (right).

## C  Experimental Setup

### C.1  Implementation Details

We consider the following representative dialogue models as baseline models to evaluate the TOP-DIAL dataset:

- **DialoGPT** (Zhang et al., 2020): It is a state-of-the-art pre-trained dialogue response generation model for multi-turn conversations. We adopt the pre-trained small[4] model (approximately 117M parameters) for fine-tuning.

- **Alpaca-7B** (Taori et al., 2023): It is an open-source instruction-following large language model (LLM), which is fine-tuned from a 7B LLaMA (Touvron et al., 2023) model. It supports diverse conversational tasks and is one of the most advanced LLMs for dialogue. To make it affordable, we fine-tune Alpaca-7B[5] on 2 NVIDIA 3090 GPUs with LoRA (Hu et al., 2022), a parameter-efficient fine-tuning approach.

Due to the much larger size of the constructed TOP-DIAL dataset compared to the seed dataset, we randomly sample 5K dialogues (close to the size of training dialogues in the seed dataset) from the

---
[4] https://huggingface.co/microsoft/DialoGPT-small
[5] https://github.com/tloen/alpaca-lora

training sets of the seed and TOPDIAL datasets, respectively. This ensures an identical data size for model training. Then, we fine-tune the above two baseline models for 5 epochs on the seed and TOPDIAL training datasets, respectively. We adopt default hyper-parameter settings for the two models based on their open-source code.

For a fair comparison, we build the test set containing 2000 samples, with 50% randomly sampled from the seed test data and 50% randomly sampled from the TOPDIAL test data. We adopt greedy search decoding for all baseline models during inference, with a maximum decoding length of 80.

### C.2  Evaluation Metrics

To evaluate the system utterance generation performance of the baseline models trained on different datasets, we adopt commonly used evaluation metrics, including the average score of BLEU-1/2 (Papineni et al., 2002), knowledge F1 (Liu et al., 2021; Wang et al., 2023a), persona F1 (Lim et al., 2022; Zhong et al., 2022), and target success rate (Succ.) (Wang et al., 2023a), following many existing studies. Concretely, the average score of BLEU-1/2 measures word overlaps of the generated utterances and the system's ground truth utterances. The knowledge F1 evaluates the performance of generating correct knowledge (e.g., topics, attributes)

```
Target: <Movie recommendation, "To Me the Way">
Domain Knowledge: <"To Me the Way", Type, "Motion Short film>, <"To Me the Way", Stars, "Gu
Tianle, Jay Chou, Edison Chan, Wu Jianhao">, ...
User Profile: {Age Range: "26-35", Name: "Xinqi Ren", Gender: "Male", Residence: "Dongguan",
Occupation": "Student", Accepted movies: "Days of Tomorrow", Accepted music: "The Promised Love;
Magician", Rejected music: "Rooftop", Accepted celebrities: "Andy Lau; Jay Chou", Rejected
movies: ""}
User Personality: {Openness: "intellectual, imaginative, and curious", Conscientiousness:
"inefficient, careless, and sloppy", Extraversion: "shy, reserved, and quiet", Agreeableness:
"trustworthy, straightforward, and generous", Neuroticism: "sensitive, nervous, and insecure"}

[system]: Hello Xinqi Ren, it's great to chat with another movie enthusiast! What are some of
your favorite movies and music artists?
[User]: Hi Yuhang Wang, I enjoy Days of Tomorrow and music from The Promised Love and Magician.
My favorite celebrities are Andy Lau and Jay Chou. I'm not a fan of Rooftop though.
[system]: I see that you're a big fan of Jay Chou. Did you know that he was named one of the 25
Most Influential People in Asia by CNN and was also included in Fast Company's Top 100 Creative
People in the World? He's achieved a lot!
[User]: Yes, I'm aware of Jay Chou's achievements. He's definitely a talented musician and I
enjoy his music. However, I tend to be more reserved and shy, so I don't really follow celebrity
news or gossip much. I just appreciate the music and movies.
[system]: That's totally understandable. Speaking of movies, have you watched To Me the Way?
It's a motion short film that has a good reputation and features Jay Chou as one of the stars.
It's definitely worth checking out!
[User]: I haven't seen To Me the Way, but I'll definitely check it out.
[system]: To Me the Way actually has a rating of 7.6, which is pretty good. Jay Chou's performance
in the movie was also recognized with a Hundred Flowers Award for Best Supporting Actor. Plus,
the movie was directed by Johnnie To, who is known for his unique style and storytelling. I
think you'll really enjoy it!
[User]: Thanks for the info, Yuhang Wang. On another note, have you watched the movie Parasite?
It won multiple awards, including Best Picture at the Oscars and I found it to be a really
interesting movie.
[system]: I have seen Parasite and I thought it was a great movie too! The plot was very unique
and the acting was superb. However, I think To Me the Way has its own unique charm and I highly
recommend giving it a watch. Let me know what you think after you've seen it!
[User]: Sure, I'll definitely let you know my thoughts after I've watched To Me the Way. Thanks
for chatting with me, Yuhang Wang. It's been great discussing movies and music with another
enthusiast!
```

Figure 9: A randomly picked curated case for personalized target-oriented dialogue.

from the domain knowledge triples. The persona F1 calculates the F1 value of the uni-grams co-occurring in the generated utterance and grounded user profile, following existing work for personalized dialogue (Lim et al., 2022; Zhong et al., 2022). The target success rate measures the proportion of correct target topic generation within the ground-truth turn and the two adjacent turns in the test set, because multiple temporary strategies can be reasonable before reaching the target due to the nature of dialogue.

## D Case Study

We provide two randomly picked cases in Figure 9 and Figure 10.

**Target**: <Movie recommendation, "Life and Death Plunder">
**Domain Knowledge**: <"Life and Death Plunder", Rating, 7.1>, <"Xun Zhou", Stars, "Life and Death Plunder", ...
**User Profile**: {Age Range: "18-25", Name: "Qiling Sun", Gender: "Male", Residence: "Nanchang", Occupation: "Employed", POI: "Going Home Hunan Restaurant", Accepted movies: "The Miracle World", Accepted music: "Chase", Accepted celebrities: "Aaron Kwok", Rejected movies: "Levin", Rejected music: "Heartburn"}
**User Personality**: {Openness: "intellectual, imaginative, and curious", Conscientiousness: "efficient, organized, and careful", Extraversion: "shy, reserved, and quiet", Agreeableness: "unreliable, complicated, meager, and boastful", Neuroticism: "secure, confident, and calm"}

[System]: Hi Qiling Sun! It's nice to chat with you. I'm also a movie enthusiast. What are some of your favorite movies?
[User]: Hi Haizheng Ma! My favorite movie is The Miracle World. It's a great adventure film that really captivated me. How about you?
[System]: That's great to hear! I really enjoyed Life and Death Plunder, a film that reflects the lives of women in modern Chinese cities. Have you seen it before?
[User]: Sorry about that, I haven't seen Life and Death Plunder yet, but it sounds interesting. As for music, I enjoy listening to Chase. And for celebrities, I really admire Aaron Kwok.
[System]: I see, Aaron Kwok is certainly a great performer! Speaking of movies, have you watched any films starring Xun Zhou? She's a talented actress and has won numerous awards for her performances. In fact, she stars in Life and Death Plunder, which I highly recommend checking out!
[User]: Oh, sorry about that. I'm actually not a big fan of Xun Zhou's movies, but I'm always open to checking out something new. Thank you for the recommendation!
[System]: Sure, no problem! Life and Death Plunder has a rating of 7.1, and Xun Zhou delivers an outstanding performance as always. Plus, it has a good reputation among critics and audiences alike. What do you think? Want to give it a try?
[User]: Thanks for the recommendation, but I think I'll pass this time. I prefer adventure and action movies like The Miracle World, but I appreciate your suggestion. Maybe I'll give it a try in the future.

Figure 10: A randomly picked curated case when the user rejects the target topic.