# OpenReview forum: "Target-oriented Proactive Dialogue Systems with Personalization: Problem Formulation and Dataset Curation"
_EMNLP/2023/Conference — EMNLP 2023 Main_

### Official Review · Reviewer_kAUS · 2023-08-05

**Soundness:** 3

**Excitement:**

3: Ambivalent: It has merits (e.g., it reports state-of-the-art results, the idea is nice), but there are key weaknesses (e.g., it describes incremental work), and it can significantly benefit from another round of revision. However, I won't object to accepting it if my co-reviewers champion it.

**Paper Topic And Main Contributions:**

The paper prsetns ad dataset TOPDIAL for the domain of target-oriented proactive dialogue systems that steer conversations toward predefined goals. The authors formulate the problem with a focus on user personalization during the target accomplishment process. They use a <dialogue act, topic> pair as the target. The authors propose an approach for dataset curation using a role-playing framework. This methodology leads to TOPDIAL, containing around 18,000 multi-turn dialogues.

**Reasons To Accept:**

- The paper addresses the emerging area of personalized target-oriented dialogue systems, which is crucial for enhancing user experience.
- The automatic dataset curation framework offers a novel and efficient approach to construct a large-scale dataset.

**Reasons To Reject:**

- The paper doesn't clearly highlight the distinctiveness of the TOPDIAL dataset compared to existing datasets like DuRecDial 2.0, particularly in terms of the added value of including personality information.
- The significance of the personalized aspect and its impact on the user experience needs to be more clearly demonstrated or discussed.
- The limitations or challenges faced during the dataset curation process, if any, are not explicitly mentioned.

**Reproducibility:**

4: Could mostly reproduce the results, but there may be some variation because of sample variance or minor variations in their interpretation of the protocol or method.

**Reviewer Confidence:**

4: Quite sure. I tried to check the important points carefully. It's unlikely, though conceivable, that I missed something that should affect my ratings.

---

> ### Author Rebuttal · Authors · 2023-08-28
>
> We sincerely thank the reviewer’s comments and constructive suggestions. Below, we address the reviewer’s concern point by point:
>
> **A1**. The comparison between TOPDIAL and existing datasets is summarized in Table 1. We highlight the distinctiveness of TOPDIAL from the following two aspects:
>
> (1) **Target-oriented proactivity**. The DuRecDial 2.0 dataset is built without consideration of targets, thus it is not exactly suitable for the end task of target-oriented proactive dialogue. The re-purposed version of DuRecDial 2.0 largely relies on human efforts to form targets and preprocess dialogues. In comparison, our TOPDIAL dataset is curated based on target-oriented proactivity. It is of higher quality in terms of desirable characteristics (see Figure 3) and serves as a more effective resource for training dialogue models for the end task (see Table 3).
>
> (2) **Personalization (particularly personality)**. For a higher user experience, target-oriented proactive dialogue systems need to consider personalization, which was largely ignored by previous work. Particularly, one’s personality may imply possible reactions and feedback during the dialogue process. Taking recommendation-related target-oriented dialogue as an example, an outgoing, straightforward, and sensitive person is generally more open to exploring new things and more likely to accept the target topic, while a shy, complicated, and calm person may stick to his/her preferences and reject the recommendation if the target topic does not fit his/her taste. Such cases require the system to take strategic dialogue acts and different speaking styles to lead the conversation towards the target. By grounding personality information during the dataset curation process, our TOPDIAL dataset is more effective for training personalized target-oriented dialogue models.
>
> In the final version, we will improve relevant contexts to highlight the TOPDIAL dataset’s distinctiveness.
>
>
> **A2**. It is crucial for target-oriented dialogue systems to consider personalized aspects, such as user profiles and personalities. The user profiles involve a user’s past preferences about potential topics relevant to the target, while personalities imply possible reactions and feedback during the dialogue process. With personalized information incorporated, the system could be tailored to a user and lead the conversation towards the target with higher engagement instead of obtrusively driving to the target, thereby improving user experience. Besides, several studies [1-2] have emphasized the importance of personalization on user experience. In the final version, we will improve the discussion in relevant contexts to make it more clearly.
>
> [1] Wu et al., Personalized Response Generation via Generative Split Memory Network. In *Proceedings of NAACL-HLT (2021)*.
>
> [2] Rana et al., User Experience and The Role of Personalization in Critiquing-Based Conversational Recommendation. *ACM Transactions on the Web (2023)*.
>
>
> **A3**. First, dataset curation **challenges** mainly lie in building clear task instructions and deciding which information should be provided to the agents as prompting sources. We conquered these challenges by following open-sourced prompting guidelines and improving our framework through preliminary human evaluations.
>
> Second, we discussed the two limitations faced during the dataset curation process in the “**Limitations**” section. One limitation is that ensuring factual correctness is difficult, which is attributed to the hallucination issue yielded by ChatGPT. We intend to address it with additional post-processing steps, such as fact-checking and correction based on the grounded domain knowledge. The other one is that, we observe that sometimes the moderator agent cannot appropriately terminate a conversation due to its difficulty in understanding target accomplishment, even though it has been assigned with detailed instructions and in-context examples. We will leave this for future research.

---

### Official Review · Reviewer_CF2z · 2023-08-10

**Soundness:** 4

**Excitement:**

4: Strong: This paper deepens the understanding of some phenomenon or lowers the barriers to an existing research direction.

**Paper Topic And Main Contributions:**

The paper proposed to automatically create a new dataset, TOPDIAL, to improve the user experience via personalization in the target-oriented dialogue system.

**Questions For The Authors:**

1. As the authors mentioned in Line 202, the user could also reject the recommendation from the system. Thus, to better understand how the recommendation is rejected by the user, could the authors provide some cases about the **rejecting situation**?

**Reasons To Accept:**

1. The paper presents a data construction method for personalized target-oriented dialogue, which is an interesting research direction.
2. The curated method is simple but effective in constructing proactive dialogue with personality.

**Reasons To Reject:**

1. I am a little concerned about the usefulness of the data construction method. Since it almost relies on large language models (e.g. ChatGPT), does it mean that ChatGPT is sufficient to solve the problem of personalized target-oriented dialogue if we provide sufficient information and design proper prompts for the LLM?

**Reproducibility:**

4: Could mostly reproduce the results, but there may be some variation because of sample variance or minor variations in their interpretation of the protocol or method.

**Reviewer Confidence:**

4: Quite sure. I tried to check the important points carefully. It's unlikely, though conceivable, that I missed something that should affect my ratings.

---

> ### Author Rebuttal · Authors · 2023-08-28
>
> We sincerely thank the reviewer’s strong support for our work. Below, we address the reviewer’s concern point by point:
>
> **A1**. We clarify that **the usefulness of dataset construction is not equal to the sufficiency of problem-solving**.
>
> First, the problem of personalized target-oriented dialogue is challenging, which requires the system to perform long-term planning to steer a conversation toward the target, and meanwhile, in a personalized manner for different users. According to our preliminary investigation, when a ChatGPT agent is provided with task instructions and sufficient information and is asked to chat directly with humans, it often fails to accomplish the target.
>
> Second, although ChatGPT is not sufficient enough to address the problem, it is helpful in simulating assigned roles. Since our user and system agents are built on top of the same LLM, their language distributions and instruction-following abilities are similar, and they are asked to interact with each other round by round, making them effective in generating desired dialogue data.
>
> Third, compared to the limitation that ChatGPT is a general conversational assistant and relies on zero/few-shot examples to address a specific dialogue task, our curated dataset is valued because it provides a wealth of high-quality data resources useful for training more effective dialogue models.
>
>
> **A2**. Below we provide two cases to show the rejecting situation:
>
> // --------------------------------- Case 1 ---------------------------------
>
> Target: < Movie recommendation, "Life and Death Plunder" >
>
> Conversation:
>
> [System]: Hi Qiling Sun! It's nice to chat with you. I'm also a movie enthusiast. What are some of your favorite movies?
>
> [User]: Hi Haizheng Ma! My favorite movie is The Miracle World. It's a great adventure film that really captivated me. How about you?
>
> [System]: That's great to hear! I really enjoyed Life and Death Plunder, a film that reflects the lives of women in modern Chinese cities. Have you seen it before?
>
> [User]: Sorry about that, I haven't seen Life and Death Plunder yet, but it sounds interesting. As for music, I enjoy listening to Chase. And for celebrities, I really admire Aaron Kwok.
>
> [System]: I see, Aaron Kwok is certainly a great performer! Speaking of movies, have you watched any films starring Xun Zhou? She's a talented actress and has won numerous awards for her performances. In fact, she stars in Life and Death Plunder, which I highly recommend checking out!
>
> [User]: **Oh, sorry about that. I'm actually not a big fan of Xun Zhou's movies**, but I'm always open to checking out something new. Thank you for the recommendation!
>
> [System]: Sure, no problem! Life and Death Plunder has a rating of 7.1, and Xun Zhou delivers an outstanding performance as always. Plus, it has a good reputation among critics and audiences alike. What do you think? Want to give it a try?
>
> [User]: Thanks for the recommendation, **but I think I'll pass this time. I prefer adventure and action movies like The Miracle World, but I appreciate your suggestion**. Maybe I'll give it a try in the future.
>
> ----------------------------------------------------------------------------
>
> // --------------------------------- Case 2 ---------------------------------
>
> Target: < Music recommendation, " Slowly" >
>
> Conversation:
>
> [System]: Hello Nalu, how are you today? I noticed that you enjoy a variety of movies and music. What are your favorites?
>
> [User]: Hi Liqi, I'm doing alright. My favorite movies are Golden Chicken 2 and Twelve Nights, and I enjoy listening to Love is Simple and Can You Feel the Love Tonight.
>
> [System]: It's great to hear that you're a fan of Jacky Cheung! Did you know that he's won the World Music Award for Best Selling Chinese Artist and the Golden Horse Award for Best Supporting Actor? He's truly a heavyweight idol.
>
> [User]: Yes, I'm aware of Jacky Cheung's achievements. I appreciate his talent and hard work. However, I'm not a big fan of discussing celebrities and their awards.
>
> [System]: Well, have you heard of the movie "Golden Chicken 2", which stars Jacky Cheung? One of my favorite songs by Jacky Cheung is "Slowly". It's a beautiful ballad that showcases his amazing vocals and versatility.
>
> [User]: Thank you for sharing your interests, **but I'm not really in the mood for discussing movies or music right now**. Is there anything else you'd like to talk about? Maybe something about your day or some interesting news you've heard recently?
>
> [System]: Sure, I can understand that. Well, speaking of music, have you ever listened to "Slowly" by Jacky Cheung? It's a wonderful song that I think you might enjoy. The comments say that he is perfect for any kinds of styles, and I couldn't agree more.
>
> [User]: I appreciate your suggestion, but **as I mentioned earlier, I'm not really interested in discussing music or movies right now**. Perhaps we can talk about something else? How about your thoughts on recent events in your city, or any interesting hobbies you have?
>
> ----------------------------------------------------------------------------
>
> In case 1, the recommendation is rejected because the target movie does not meet the user’s preferences. In case 2, it is because the user expresses that he/she is not in the mood for discussing target-related topics. We will add detailed case studies in the final version.

---

### Official Review · Reviewer_4n1y · 2023-08-11

**Soundness:** 4

**Excitement:**

4: Strong: This paper deepens the understanding of some phenomenon or lowers the barriers to an existing research direction.

**Paper Topic And Main Contributions:**

The paper is about formulating a <dialogue act, topic> pair as the conversation target, we explore a novel problem of how to improve user experience by considering their personalization during the target accomplishment process.

Contributions: new data resources (particularly for low-resource languages)

**Reasons To Accept:**

A good paper with new dataset. It is new dataset which is good.

**Reasons To Reject:**

Good paper to accept.

**Reproducibility:**

3: Could reproduce the results with some difficulty. The settings of parameters are underspecified or subjectively determined; the training/evaluation data are not widely available.

**Reviewer Confidence:**

2: Willing to defend my evaluation, but it is fairly likely that I missed some details, didn't understand some central points, or can't be sure about the novelty of the work.

---

> ### Author Rebuttal · Authors · 2023-08-28
>
> We sincerely thank the reviewer’s strong support for our work. Our contributed TOPDIAL dataset is the first dataset equipped with the desirable characteristics for personalized target-oriented dialogue. It is new, high-quality, and large-scale, significantly boosting the exciting research field.

---

### Meta-Review · Area_Chair_7Umo · 2023-09-19

**Recommendation:** 4

**Metareview:**

The paper proposed to automatically create a new dataset, TOPDIAL, to improve the user experience via personalization in the target-oriented dialogue system. The data construction method is simple but effective. The dataset can be a good resource for the dialogue community.

The paper can be improved by clearly 1) highlighting the distinctiveness of the TOPDIAL dataset compared to existing datasets like DuRecDial 2.0 and 2) discussing the significance of the personalized aspect and its impact on the user experience.

---

### Decision · Program_Chairs · 2023-10-07

**Decision:**

Accept-Main

**Comment:**

The paper proposed to automatically create a new dataset, TOPDIAL, to improve the user experience via personalization in the target-oriented dialogue system. The data construction method is simple but effective. The dataset can be a good resource for the dialogue community.

The paper can be improved by clearly 1) highlighting the distinctiveness of the TOPDIAL dataset compared to existing datasets like DuRecDial 2.0 and 2) discussing the significance of the personalized aspect and its impact on the user experience.